# Enhancing Clinical Reasoning with Virtual Patients: A Hybrid Systematic Review Combining Human Reviewers and ChatGPT

**DOI:** 10.3390/healthcare12222241

**Published:** 2024-11-11

**Authors:** Daniel García-Torres, María Asunción Vicente Ripoll, César Fernández Peris, José Joaquín Mira Solves

**Affiliations:** 1ATENEA Research Foundation for the Promotion of Health and Biomedical Research of Valencia Region (FISABIO), 03550 Sant Joan d’Alacant, Spain; daniel.garciat@umh.es (D.G.-T.); jose.mira@umh.es (J.J.M.S.); 2Telematics Engineering Area, Miguel Hernández University of Elche, 03202 Elche, Spain; c.fernandez@umh.es; 3Health Psychology Department, Miguel Hernández University of Elche, 03202 Elche, Spain; 4Health District Alicante-Sant Joan, 03013 Alicante, Spain

**Keywords:** virtual patients, medical education, clinical reasoning, ChatGPT

## Abstract

Objectives: This study presents a systematic review aimed at evaluating the effectiveness of virtual patients in enhancing clinical reasoning skills in medical education. A hybrid methodology was used, combining human reviewers and ChatGPT to assess the impact of conversational virtual patients on student learning outcomes and satisfaction. Methods: Various studies involving conversational virtual patients were analyzed to determine the effect of these digital tools on clinical competencies. The hybrid review process incorporated both human assessments and AI-driven reviews, allowing a comparison of accuracy between the two approaches. Results: Consistent with previous systematic reviews, our findings suggest that conversational virtual patients can improve clinical competencies, particularly in history-taking and clinical reasoning. Regarding student feedback, satisfaction tends to be higher when virtual patients’ interactions are more realistic, often due to the use of artificial intelligence (AI) and natural language processing (NLP) in the simulators. Furthermore, the study compares the accuracy of AI-driven reviews with human assessments, revealing comparable results. Conclusions: This research highlights AI’s potential to complement human expertise in academic evaluations, contributing to more efficient and consistent systematic reviews in rapidly evolving educational fields.

## 1. Introduction

### 1.1. Objective and Scope of This Article

The primary objective of this article is to conduct a systematic review of the literature on the use of virtual patients in medical education. Virtual patients, which are computer-based programs simulating real-life clinical scenarios, represent a significant advancement in digital education aimed at enhancing clinical reasoning skills among medical students. This review aims to evaluate the effectiveness of these tools in improving diagnostic accuracy and clinical reasoning, as well as to provide a comprehensive analysis of the latest trends and developments in the use of virtual patients in medical education.

In addition to assessing the current state of research on virtual patients, this review seeks to validate the performance of ChatGPT-4 [1] as a reviewer in comparison to human experts. The use of ChatGPT-4 in the review process offers an innovative approach to handling large volumes of academic literature, providing consistent and objective evaluations, and potentially accelerating the review process [2,3].

By combining human expertise with the analytical capabilities of ChatGPT-4, we aim to determine whether AI can match or even enhance the quality of systematic reviews traditionally conducted by human reviewers.

Our work also aimed at analyzing the evolution and future prospects in virtual patients, which, thanks to technological advancements, are becoming increasingly realistic. We analyzed which technologies have contributed to this process and which are most relevant today.

Ultimately, this article contributes to the growing body of knowledge on digital education tools in healthcare while exploring the potential of AI to support and enhance academic research processes. This dual focus on educational innovation and methodological advancement underscores the significance of integrating AI into systematic reviews to keep pace with rapid technological advancements in medical training.

### 1.2. Importance and Challenges of Teaching Clinical Reasoning in Medical Education

Clinical reasoning, encompassing the cognitive processes underlying medical decision-making, is a fundamental competency in clinical practice, serving as the bedrock for diagnostic and therapeutic actions [4]. Hence, it is imperative to train medical students not only in technical knowledge but also in clinical reasoning and critical reflection skills to adeptly navigate the complexity and ambiguity inherent in clinical practice [5].

Despite its significance, clinical reasoning is often explicitly taught to a limited extent within medical curricula [6]. This shortfall persists even though experts advocate for its integration at all stages of medical education, citing barriers such as curriculum time constraints and insufficient faculty expertise [7]. As a result, newly trained physicians frequently report feeling inadequately prepared for clinical reasoning and diagnosis [8].

Digital education offers a potential solution to these challenges, utilizing various tools designed to enhance learning and clinical reasoning skills. A focal point of our research is the employment of virtual patients. Virtual patients are computer-based programs that simulate real-life clinical scenarios, enabling students to practice decision-making in a controlled, interactive environment [9]. A virtual patient must simulate the behavior of real patients as well as the evolution of their health state, which can be modeled using, for example, Petri nets, as proposed in [10]. Additionally, these simulations can incorporate complex case data, provide immediate feedback, and present diverse clinical contexts, thereby serving as a powerful educational resource to improve diagnostic accuracy and clinical reasoning skills [11,12].

Since the early 2000s, virtual patients have been employed in medical education to address escalating training expectations amid dwindling training resources [12]. The interactivity of virtual patients ranges from basic textual simulations to intricate environments featuring multimedia elements such as images and sounds [13]. More recently, the advent of AI has facilitated conversational simulations, allowing students to interact with patients in real-time [14,15]. These tools not only enhance students’ clinical skills but also offer a safe space to make mistakes and learn from them without endangering actual patients [14].

Although there are several reviews on this topic [16], many focus on communication skills and do not adequately address the effectiveness of virtual patients in improving clinical reasoning skills. Additionally, rapid technological advancements necessitate the inclusion of recent studies in these reviews. For instance, recent reviews [17] explore AI and machine learning-based systems, which were not considered in earlier reviews.

### 1.3. Use of ChatGPT or Other LLMs as Article Reviewers

In recent years, artificial intelligence (AI) has emerged as a valuable tool in various fields of academic and medical research. Large language models (LLMs), such as ChatGPT developed by OpenAI, exemplify how AI can contribute to the systematic review process of scientific articles. These models, particularly those based on the GPT-4 architecture, possess the capability to analyze large volumes of text, identify patterns, and provide detailed feedback in significantly less time than human reviewers. For instance, in the work of Katz et al. [18], the remarkable advancement of language models, particularly GPT-4, is highlighted. This research evaluates GPT-4’s performance on the Bar Exam, including the multiple-choice, essay, and practical tests. GPT-4 significantly outperformed its predecessors and even achieved higher scores than the average human examinee in several areas.

The use of LLMs like ChatGPT as article reviewers offers several advantages. First, AI can efficiently handle repetitive tasks and data analysis, allowing human reviewers to focus on more complex and critical aspects of the review process. Second, ChatGPT can assist in identifying grammatical errors, coherence issues, and other technical aspects of the manuscript, thereby enhancing the overall quality of the text prior to final human review.

Moreover, integrating ChatGPT into the review process ensures greater consistency and objectivity in manuscript evaluation. AI can apply the same evaluation criteria uniformly, reducing potential bias that may arise from the differing perceptions and experiences of human reviewers. This not only improves the quality of the review but also expedites the process, which is particularly valuable in rapidly evolving fields such as medical education and health technologies [19].

It is crucial to acknowledge, however, that the role of ChatGPT should be complementary to that of human reviewers. While AI can provide valuable preliminary analysis, the expertise and clinical judgment of human experts remain irreplaceable for assessing the scientific relevance, innovation, and clinical applicability of articles. Therefore, a collaborative approach that combines the strengths of AI and human review can yield the best results in the evaluation of scientific literature [20].

In this context, our article employs a combination of human reviewers and ChatGPT, specifically GPT-4 (released on 14 March 2023) and GPT-4o (released on 13 May 2024), to conduct a systematic review and technological trend analysis on the use of virtual patients in medical education. This hybrid methodology aims to leverage the strengths of both approaches, ensuring a thorough and rigorous review of the available literature.

## 2. Materials and Methods

This review adheres to the PRISMA (Preferred Reporting Items for Systematic Reviews and Meta-Analyses) guidelines [21]. Additionally, it has been registered in the PROSPERO International Prospective Register of Systematic Reviews under registration number CRD42024574334, ensuring transparency and adherence to established protocols for systematic reviews.

### 2.1. Eligibility Criteria

Participants in the included studies were required to be enrolled in a health education or training program. This included students from disciplines such as medicine, dentistry, nursing and midwifery, medical diagnostic and treatment technology, physiotherapy and rehabilitation, or pharmacy.

This review focused on studies where the virtual patient was conversational, either by voice or by keyboard, using natural language in all cases, and where virtual patient-based training was compared to a control group or an alternative method.

Articles were not considered eligible if the goal of the training was not to diagnose correctly or to determine the correct treatment for a patient. Additionally, duplicate entries, papers not fulfilling the inclusion criteria according to the title and abstract, and papers not fulfilling the inclusion criteria according to their full texts were also excluded.

The primary eligible outcomes were learning impact, represented as clinical competencies measured post-intervention with validated or non-validated instruments, and student satisfaction, measured with satisfaction surveys. However, studies that do not directly measure learning impact but include a conversational model related to education were also considered.

### 2.2. Search Strategy

Concerning the systematic review, four databases were used for the search: PubMed, EMBASE, Scopus, and ProQuest. This combination of specialized databases (PubMed and EMBASE) and interdisciplinary databases (Scopus and ProQuest) allowed us to cover the most relevant studies in medical and healthcare research, as well as additional studies related to virtual patients that could fit in technology or other non-medical journals. ProQuest also allowed us to perform automated technological trend analysis through full text processing.

Database searches were performed in April 2024 with the following key search query: (“virtual patient” OR “virtual standardized patient”) AND (“diagnose” OR “clinical reasoning”) in title, abstract, and keywords. The search was limited to studies published from 1998 (according to our searches, this is the first appearance of the term “virtual patient”), in any language. The reference lists of other systematic reviews previously identified as relevant to the study topic were also examined. This approach allowed the identification of additional studies that might not have been retrieved in the initial search due to potential limitations of the databases or the search terms used.

### 2.3. Data Collection and Screening

The search results were consolidated into a single Microsoft Excel 365 document. Screening was carried out by two independent human reviewers who assessed the titles and abstracts to identify potentially eligible studies, which then progressed to a second filtering phase. Full-text articles were retrieved and evaluated based on predefined eligibility criteria. Any discrepancies were resolved through discussion between both reviewers. AI tools were not used in the screening phase.

The PRISMA flow diagram presented in Figure 1 outlines the process. A total of 486 records were identified: 192 from Scopus, 137 from EMBASE, 90 from PUBMED, and 67 from PROQUEST. After removing 242 duplicate records, 246 records remained for screening. Of these, 174 records were excluded after abstract screening as they did not meet the inclusion criteria. The remaining 72 records were assessed for eligibility, and 62 reports were excluded due to being non-conversational VPs. A further quality assessment was carried out with those 10 remaining studies. Since all were non-randomized studies, the ROBINS-I tool (2016) [22] was applied, resulting in the following risks of bias rating: two studies were categorized as low/moderate-risk studies, five as moderate-risk studies, and three as moderate/high-risk studies. No high-risk or critical-risk studies were found, and thus all were deemed suitable for inclusion in the systematic review.

### 2.4. Data Extraction by Human Reviewers and GPTs

The remaining studies kept after screening were selected for data extraction. For this purpose, we engaged two human reviewers (HR1 and HR2) and two versions of the artificial intelligence ChatGPT (GPT-4 and GPT-4o) configured for review tasks.

Human reviewers HR1 and HR2 worked in parallel and stored their results in independent Excel documents. In parallel as well, a third human reviewer, HR3, was in charge of carrying out information extraction using AI.

Concerning information extraction using AI, to address potential variability in ChatGPT’s responses, each extraction prompt was conducted three times in independent, parallel conversations. The initial prompts used for information extraction are provided in Appendix A.

Subsequently, we consolidated the three resulting tables (Appendix A) to create a single results table per article. Our method ensured that no information was omitted from the search and no false information was generated or included in the articles. We extracted relevant information independently for each included study, covering: the software/AI used for creating the virtual patient, the number of participants and their division into control and other groups, the specialty or hospital area to which the students belonged, the impact on student learning, and student satisfaction. All relevant data were extracted using a structured form in Microsoft Excel, like the forms used by the human reviewers HR1 and HR2.

Once all results were available (HR1, HR2, GPT-4, and GPT-4o), the third human reviewer (HR3) compared all responses. In cases of discrepancies, HR3 selected the most voted answer or, if there was an equal number of votes for multiple options, made a final decision independently.

After all the final answers were determined, it was possible to rate the reviewers, verifying whether each reviewer correctly identified each of the following categories in every article: Software/AI used, Software description, Participants, Control and other groups, Hospital Area, Learning Impact, and Student Satisfaction.

Frequency tables were used for counts. The accuracy in each category was compared using the chi-square test for each pair of reviewers.

### 2.5. Technological Trend Analysis

To analyze the evolution of technologies involved in virtual patients and forecast future prospects, we conducted an additional experiment using only the ProQuest database (as this tool allows for the automated extraction of full texts from all retrieved articles). We relaxed the search criteria to increase the number of relevant works, retaining only “virtual patient” OR “virtual standardized patient” as the search terms, without mentioning “diagnose” OR “clinical reasoning”. On the full texts of these articles, we measured the frequency of several terms representative of the most influential technologies in virtual patient creation. Specifically: “head-mounted display” (or its abbreviation “HMD”); “3D”; “virtual reality” (or its abbreviation “VR”); “avatar”; “natural language processing” (or its abbreviation “NLP”); and “artificial intelligence” (or its abbreviation “AI”).

## 3. Results

This section presents the detailed results of the literature review on the use of virtual patients in medical education, focusing primarily on the aspects of “Software/AI Used”, “Learning Impact”, and “Student Satisfaction”. Finally, the comparison between human reviewers and GPTs will be discussed.

Table 1 below summarizes the main results obtained from the literature review. This table provides an overview of the analyzed studies, highlighting the software used, the number of participants, the impact on learning, and student satisfaction.

### 3.1. Software Used

Among the reviewed studies, we found a wide variety of software utilized. For instance, Kleinheksel et al. [25], Isaza-Restrepo et al. [27], and Kamath et al. [31] used web-based platforms that create interactive clinical scenarios and simulate patient interactions through chat interfaces and virtual tools. In contrast, Graham et al. [29] utilized the Virtual Human Toolkit, which combines multiple functionalities: apart from student-patient conversations, there are nurse-doctor conversations, availability of clinical records, and videos showing the patient behavior depending on the actions chosen.

More advanced technologies were also used. Lin et al. [24] incorporated NERVE (Neurological Examination Rehearsal Virtual Environment), which leverages voice recognition and virtual controllers. Maicher et al. [26] employed ChatScript and Unity 3D to manage conversations between students and virtual patients through natural language processing (NLP) in 3D environments. The Julia chatbot developed by Suarez et al. [30] employed Dialogflow for AI-powered conversational flows, while Wang et al. [28] also utilized AI and NLP to simulate real-time clinical encounters. Finally, Yadav et al. [32] integrated virtual reality via Oculus Quest, combining modeling, animation, and patient interaction.

### 3.2. Learning Impact

The reviewed studies indicate that the use of virtual simulations has a significant impact on various clinical competencies among students.

The most frequently reported learning impact is the enhancement of clinical reasoning or diagnostic skills, mentioned in five of the 10 selected papers [23,26,27,31,32] (see Table 1). However, these studies present notable differences. For example, Maicher et al. [26] suggest that, for improvements in clinical reasoning, virtual patients should simulate complex clinical cases that challenge students and be tailored to their experience level (i.e., first-year students vs. third-year students). Less experienced students may ask broader questions, making the virtual patient’s responses less predictable. Isaza-Restrepo et al. [27] provide the only study with quantitative data on learning impact, showing statistically significant improvements in clinical reasoning through a pre-post analysis. Courteille et al. [23] focus on OSCE exams using virtual patients, where clinical reasoning is one of the key points to evaluate. Kamath & Ullal [31] conclude that, after completing a case scenario, students are better prepared to handle similar cases. In Yadav et al. [32], immersive and realistic learning experiences facilitated not only technical skills practice but also the development of more structured and logical clinical thinking.

Improvements in history-taking skills are also commonly reported, appearing in three of the 10 selected papers [24,26,27]. Isaza-Restrepo et al. [27] again provide quantitative data, demonstrating significant improvement in interview skills in a pre-post study. In the study by Lin et al. [24], the interview seemed better suited to third- and fourth-year students than to first- and second-year students, highlighting the need for virtual patients to be adapted to students’ experience levels—a point also noted by Maicher et al. [26].

Other relevant learning impacts include improvements in communication skills and confidence, each mentioned in two studies. Specifically, communication skills are highlighted in the studies by Suárez et al. [30] and Kamath & Ullal [31], while confidence improvements are noted by Suárez et al. [30] and Wang et al. [28]. The main characteristic of the virtual patient presented in Suárez et al. [30] is its simplicity, since it was created as an AI chatbot using general purpose tools. Despite this, it allowed an improvement in communication skills and confidence, which emphasized the power of AI to build simple, yet useful, virtual patients. In the work by Kamath & Ullal [31], apart from the improvements in communication skills, a specific emphasis is given to the ethical and behavioral aspects of patient care.

A recurring theme in four studies is the adaptation of virtual patients based on students’ experience levels. Maicher et al. [26] and Wang et al. [28] suggest that virtual patients are best suited for early-stage students or novices, while simulated patients (i.e., actors) may offer additional benefits for more advanced students. Conversely, Lin et al. [24] present a differing result, finding that the virtual patient approach was more suitable for third- and fourth-year students, as previously noted.

### 3.3. Student Satisfaction

Overall, students reported high levels of satisfaction with simulators that utilized advanced technologies such as AI and NLP. In studies like those by Suárez et al. [30] and Wang et al. [28], students appreciated interacting with chatbots and virtual patients that provided real-time feedback, which increased their confidence and clinical competence.

Virtual reality also received positive evaluations. In the study by Yadav et al. [32], students found that the Oculus Quest platform improved their understanding of clinical concepts and was easy to use once they became familiar with the system. The immersive experience provided by virtual reality was particularly well received, enhancing both clinical reasoning and decision-making skills. Similarly, interactive software like that shown by Lin et al. [24] reported improvements in students’ examination and clinical history-taking skills. These simulators allowed for repetitive practice in a standardized environment, which was seen as beneficial for their training. This sense of security was also reported in studies by Isaza-Restrepo et al. [27] and Kamath et al. [31].

However, not all experiences were uniformly positive. For example, in the study by Courteille et al. [23], while most students found the virtual patient case realistic and engaging, some reported frustration with the system’s limitations, particularly in the interactive dialogue with the patient.

### 3.4. Comparison Between Human Reviewers and GPTs

The results of the comparison between human reviewers (HR1 and HR2) and the artificial intelligence models (GPT-4 and GPT-4o) demonstrate that both AI systems achieved a high level of accuracy in identifying key categories within the scientific articles.

The results indicate that GPT-4 achieved 91.4% correct responses (i.e., most voted responses) with an error rate of 8.6%. GPT-4o demonstrated higher accuracy, with 97.1% correct responses and a 2.9% error rate. Human reviewers HR1 and HR2 achieved 95.7% and 97.1% correct responses, with error rates of 4.3% and 2.9%, respectively, as shown in Figure 2. Although GPT-4 displayed a higher error rate across different categories, the statistical tests presented in Table 2 indicated no significant differences between the human reviewers and the AI models.

GPT-4o, in particular, demonstrated performance comparable to or even surpassing that of the human reviewers across several categories, as shown in Figure 3. In this figure, each category represents a key area assessed during the review process, including “Control and other groups”, “Hospital Area”, “Learning Impact”, “Participants”, “Software description”, “Software/AI use”, and “Student Satisfaction”. GPT-4o achieved the highest accuracy in most categories, especially excelling in “Software description”, where it outperformed both human reviewers. This suggests a higher level of precision in identifying specific software-related details.

On the other hand, GPT-4, while generally accurate, displayed a slightly lower performance in certain categories. It was the only evaluator to make multiple errors within a single category, specifically in ‘Software description’ and ‘Control and other groups’. This indicates that GPT-4 may have limitations in consistently identifying details within these complex categories, compared to both GPT-4o and human reviewers.

### 3.5. Technological Trend Analysis Results

Using the broader search described in the methodology section, ProQuest returned 557 articles related to virtual patients, which were analyzed to perform a count of the appearances of all technology-related terms proposed. The results obtained are displayed in Figure 4. Such results show some interesting facts. First, as expected, the term VR is present from the first studies, and its appearance frequency has kept increasing. Another interesting evolution is that of the term “avatar”, which first appeared in 2009 and is progressively gaining incidence. On the other hand, terms like HMD or NLP should not be considered relevant trends. Finally, we can see that the initial studies on the use of virtual patients were rarely associated with AI techniques. However, from 2010 on, the term appears with increasing frequency. Due to the recent advances in AI (particularly in conversational chatbots like Open AI’s ChatGPT [1]), an even higher increase in the relevance of this term is expected for the following months and years. Globally, according to the dot clouds, the combination of AI, 3D, and avatars seems to reflect the current priorities for virtual patient development.

## 4. Discussion

The total number of studies selected for review (10 studies) is lower than expected, despite the potential interest in conversational virtual patients for medical and health education. The most likely cause is the novelty of the technologies enabling the creation of these virtual patients. An increase in studies on this topic is expected in the coming years, thanks to advances in AI and the relative ease with which a conversational virtual patient can currently be created. It would be beneficial to conduct a new search in approximately one year to confirm the predicted trend.

Additionally, conversational VPs are, nowadays, even more useful for training medicine students due to the increase in the use of teleconsultations. A teleconsultation is almost perfectly simulated with a conversational VP, since the interaction is in both cases through a computer screen. The use of teleconsultation has increased significantly since the pandemic. According to the report presented in a recent analysis [33], teleconsultation usage in 2023 has been five times that of 2019, and it is expected to keep increasing in the following years. In this sense, conversational VPs should be considered a highly valuable tool for both current and future medical practice.

Concerning comparisons with other reviews, there are not many previous reviews related to conversational virtual patients, making it difficult to draw direct comparisons with the results of our study. We can only mention two: first, the one presented in Milne-Ives et al. [14], which is specific to training of communication and counseling skills for pharmacy students and pharmacists. Such study identifies four aspects in which VPs provide improvements: knowledge and skills, confidence, engagement with learning, and satisfaction, two of which are precisely the outcomes sought in our research (knowledge/skills and satisfaction). Although the studies analyzed in [14] show improvements in the four previous aspects, the authors note that many studies were small-scale without robust findings, and, consequently, further quality research was required. The second previous review, presented in Richardson et al. [16], analyzes a total of 12 studies, but only six are related to conversational virtual patients since the rest analyze other uses of AI and Machine Learning. Among the six studies related to our work, the role of linguistic authenticity in the context of communication skills is emphasized. This result aligns with our selection of studies, in which we exclusively considered conversational virtual patients using natural language communication and excluded semi-conversational environments (typically involving option selection from lists rather than using natural language), as they offer less authenticity.

Regarding the review process itself, AI has demonstrated accuracy comparable to that of human reviewers in extracting information from articles (Table 2), with GPT-4o achieving a success rate equal to or greater than human reviewers. It is reasonable to assume that in the near future, AI will be capable of conducting systematic reviews more reliably than humans; the question is how close that future is. Given the pace at which AI is improving in various tasks, this timeframe is likely measured in months rather than years. The progress is so rapid that benchmarks or comparative tests of AI capabilities against human abilities quickly become outdated. This phenomenon is illustrated in the study presented in Kiela et al. [34], which shows the necessity of proposing increasingly complex tests, and in all cases, AIs rapidly achieve human-level performance on these new tests.

This improvement speed is also evident in Katz et al. [18], where it is shown that in 2023 (with OpenAI’s GPT-4), AIs had already surpassed the average human performance in law exams. This is a remarkable improvement, considering that just one year earlier, in 2022, their results were generally worse than random guessing.

The speed in the improvement of AI capabilities is also patent on our hybrid review results, which demonstrate that GPT-4.o clearly outperforms GPT-4. The differences between GPT-4 and GPT-4.o are due to their respective stages in the evolution of large language models (LLMs). While GPT-4 is known for generating coherent and contextually accurate text across various topics, GPT-4.o introduces several improvements, including enhanced accuracy, faster processing speeds, and a greater ability to handle complex language. Additionally, GPT-4.o benefits from more extensive training data and refined algorithms, making it a superior tool for specialized tasks such as medical education and systematic reviews.

In this context, there are already studies proposing the use of AI to enhance systematic review processes, such as the one presented in Fabiano et al. [35]. Additionally, several tools are being introduced that enable semi-automated reviews. A recent compilation and analysis of tools can be found in the work by Bolaños et al. [36].

## 5. Conclusions

This study demonstrates the significant potential of virtual patients, particularly those enhanced by AI, in improving clinical reasoning skills within medical education. The findings highlight the growing importance of integrating advanced digital tools into educational practices to enhance learning outcomes. By comparing human and AI-driven reviews, we found that AI tools like GPT-4 and GPT-4.o can effectively complement human expertise, offering consistent and accurate evaluations. Future research should continue exploring AI’s role in refining and optimizing systematic reviews and educational assessments.

## Figures and Tables

**Figure 1 healthcare-12-02241-f001:**
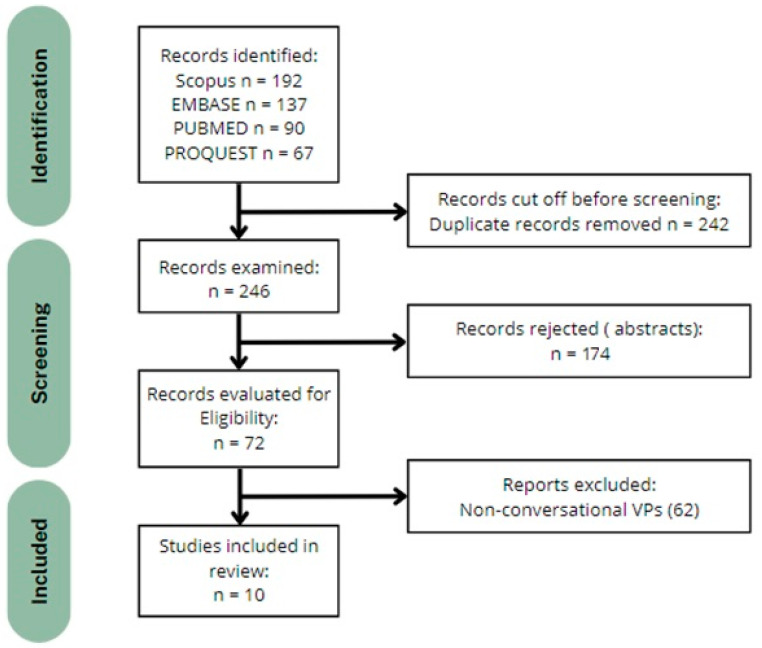
Flow diagram of the studies through the revision process.

**Figure 2 healthcare-12-02241-f002:**
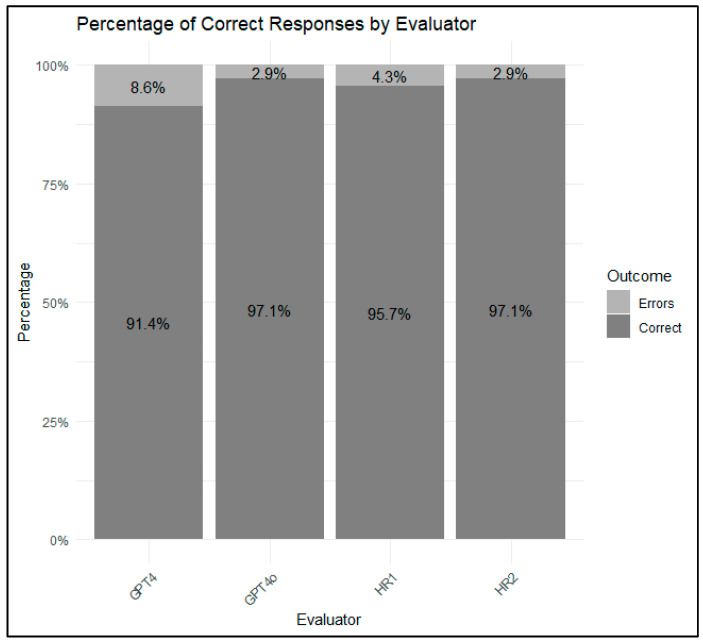
Percentage of Correct Responses (i.e., most voted responses) by Evaluator.

**Figure 3 healthcare-12-02241-f003:**
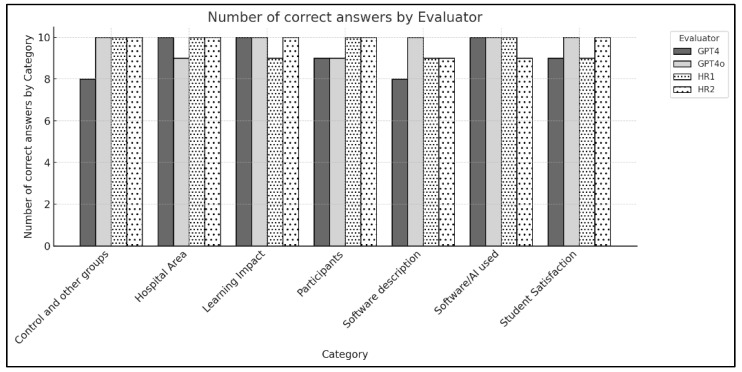
Accuracy of Human Reviewers and AI Models in Identifying Key Categories.

**Figure 4 healthcare-12-02241-f004:**
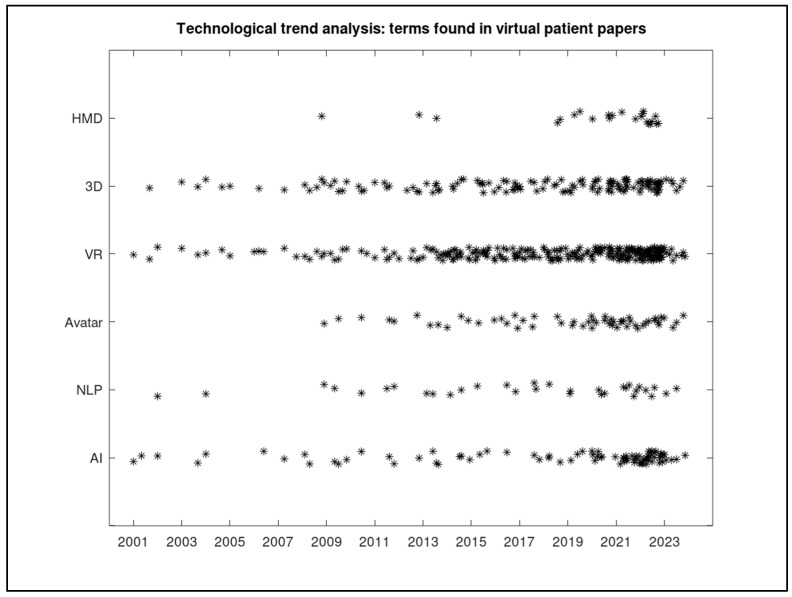
Technological trend analysis in papers related to virtual patients or virtual standardized patients. Each * represents a publication where the searched term appears. A jitter effect has been added to improve visualization.

**Table 1 healthcare-12-02241-t001:** Results of the literature review ordered chronologically.

References	Software Used	Number of Patients/Groups/Hospital Area	Learning Impact	Student Satisfaction
Courteille et al. [23] (2008)	ISP (Interactive Simulation of Patients)	110 medical students (4th year), Medicine	Enhanced clinical reasoning	Mixed reviews on usability and examination tool potential
Lin et al. [24] (2012)	NERVE: Neurological Examination Rehearsal Virtual Environment	69 (9 clinicians, 7 residents, 53 students), Neurology	Improved exam techniques and history-taking skills.	General high ratings from surveys
Kleinheksel [25] (2014)	Shadow Health Digital Clinical Experience (DCE)	130 students (Master of Science in nursing), Nursery	Achievement of deeper learning and self-reflection through long interactions with the virtual patient.	Not measured
Maicher et al. [26] (2017)	ChatScript for conversation; Unity for animation	141 students. 12,000 questions asked to the virtual patient, Medicine	Improved history-taking and diagnostic skills, particularly in early practice.	Not measured
Isaza-Restrepo et al. [27] (2018)	Own software: “the virtual patient: simulation of clinical cases”	20 medical students, Medicine	Significant improvement in history taking and clinical reasoning	Found it easy-to-use and motivating
Wang et al. [28] (2020)	Not mentioned.	112 medical students, Medicine	Increased confidence and proficiency, particularly in novices.	Increased confidence and proficiency
Graham et al. [29] (2022)	Virtual Human Toolkit (Institute of Creative Technologies, University of Southern California)	274 medical and surgical nurses, Nursery	It helped to identify gaps in pain recognition and treatment practices.	Not measured
Suárez et al. [30] (2022)	Dialogflow application.	193 students of 4th or 5th year of dentistry, Dentistry	Improved communication skills and confidence	Generally positive, especially interaction satisfaction
Kamath & Ullal [31] (2023)	OpenLabyrinth (v3.4)	20 students and 12 teachers, but one student did not provide feedback, so total 31, pharmacology	Improved communication skills and clinical reasoning	Positive feedback on real-life decision-making simulation
Yadav et al. [32] (2023)	Unity and Oculus Quest.	113 in total: 98 students and 15 faculty members. But 7 students did not fill out the questionnaires (so a real total of 106), Physiotherapy	Enhanced clinical reasoning and decision-making skills	Generally positive, especially on concept understanding and ease of use

**Table 2 healthcare-12-02241-t002:** Statistical Comparison of Accuracy Between Human Reviewers and AI Models.

Comparison	Chi Square	*p*-Value
HR1–HR2	0.15	1.00
HR1–GPT4	0.32	0.99
HR1–GPT4o	0.26	1.00
HR2–GPT4	0.32	0.99
HR2–GPT4o	0.21	1.00
GPT4–GPT4o	0.43	0.99

## Data Availability

Where appropriate, additional data or information will be shared upon receiving a reasonable request.

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
