# Peer review of "Enhancing Clinical Reasoning with Virtual Patients: A Hybrid Systematic Review Combining Human Reviewers and ChatGPT"

_healthcare, 2024, doi:10.3390/healthcare12222241_

Round 1

Reviewer 1 Report

Comments and Suggestions for Authors

Minor edits required see comments in the file attached, recommend for publication

Reviewer 2 Report

Comments and Suggestions for Authors

No specific comments to be delivered.

Only one questions related to the processing of the selected articles: where do the three cited separated conversations come from? Was that the same process repeated three times? Or does it refer to HR1, HR2, and the third reviewer? Or what else?

For what concerns the clinical reasoning, a paper to be considered is the following: Ricci, F. L., Consorti, F., Pecoraro, F., Luzi, D., & Tamburis, O. (2022). A petri-net-based approach for enhancing clinical reasoning in medical education. IEEE Transactions on Learning Technologies, 15(2), 167-178.

Reviewer 3 Report

Comments and Suggestions for Authors

This article describes a systematic review of the literature assessing the use of virtual patients to improve clinical reasoning skills. The authors use an interesting approach, combining traditional review methods utilizing human reviewers with reviews conducted using artificial intelligence (AI). The article is well-written and flows relatively well but is missing important details as outlined below.

Abstract:

·         Clear and complete. No suggestions for revision.

Introduction:

·         Informative and well-structured. No suggestions for revision.

Materials and Methods:

·         The first sentence in the Eligibility Criteria section does not seem to align with the rest of the methods or the results. “We excluded review articles and all trials related to fields other than medicine or nursing” implies that only those studies in medical and nursing students would be included. But the following sentences state that articles including any students enrolled in a health education or training program would be eligible, including “medicine, dentistry, nursing and midwifery, medical diagnostic and treatment technology, physiotherapy and rehabilitation, and pharmacy.” The results include studies in pharmacology and physiotherapy, so I assume that these disciplines were included. Consider restating the first sentence to align with the approach.

·         A strength of this study is the use of AI and comparison of these results to those of human reviewers. However, the description of the procedure for using AI (last two paragraphs in Materials and Methods) is unclear and needs more detail. AI has been used to conduct systematic reviews in both the screening and data extraction phases (see Bolanos et al, 2024). It seems, though I am not certain, that AI was used only in the data extraction phase in this study. A rationale for using it in this manner, and not in the screening process, should be included.

·         Related, the first sentence on page 5 (line 182) states that, “This reviewer uploaded the PDF versions of the scientific articles to ChatGPT.” Based on the results presented, I assume that only the 10 articles that were retained were uploaded, but this is not immediately apparent.

·         Given that both HR1 and HR2 did not show 100% accuracy, it is presumed that answers were identified as “correct” by another source. On what basis was the “correct” answer identified? Was this determined by a human reviewer or someone else? Because a large portion of the results depend on this methodology, more detail is needed for the reader to fully understand the approach to measuring accuracy.

Results:

·         The authors state that one study [21] utilized videos of patient interactions, rather than simulated patient interactions between the learner and a virtual patient. This does not seem to align with the inclusion criteria stated in the Materials and Methods (page 3, lines 130-132.)

·         When Courteille et al was mentioned in the Learning Impact section as a study that utilized high-fidelity interactive simulations, I looked at the previous section (Software Used) to read about the software for this particular study and realized that it was not included in the description. All other retained studies were included. Consider adding it to this section for completeness of reporting.

·         Given the focus of the systematic review on learning, it would be interesting to know how learning was measured in each study. The Learning Impact section could be improved by the addition of more details related to measurement.

Discussion:

·         The authors present a well-written summary and discussion of the results of the systematic review, including a discussion of their findings related to accuracy of AI compared to human reviewers. The inclusion of a new set of methods and results starting on page 10, line 358 is confusing.  While interesting, it does not seem directly related to the objective of the study as stated.  I would suggest either making it part of the study methodology and add details in the Materials and Methods and Results or moving this to the Introduction section.

·         Best practice for systematic reviews is to include some type of quality assessment to assess risk of bias. Was a quality assessment conducted on the retained articles?

Tables

·         It would be helpful for the reader to include publication dates for each study in Table 1, particularly given the focus on technologies that are constantly evolving.

·         The rationale for the ordering of the studies in the table is unclear. Perhaps by adding dates, the table can be ordered from least recent to most recent. Or the table could be arranged alphabetically by author.

Figures

·         The color scheme used in Figure 3 is problematic in regard to accessibility. Using blue and green side-by-side, for example, makes it difficult for readers with color blindness to differentiate between the two. Further, I am not sure what the figure adds to the paper in its current form. This type of graph allows visual comparison for the bottom section. After that, it is very difficult to tell, visually, where the differences lie. Consider using a different type of figure or removing this one altogether.

References

·         Citations are complete and up-to-date.

Round 2

Reviewer 3 Report

Comments and Suggestions for Authors Thank you for the opportunity to review the revised paper. I found the revisions to be very responsive to the suggestions made, and I think the strength of the manuscript has been increased significantly. It is clear that the authors put a lot of work into improving the paper, and I believe the paper now warrants publication.
I do have one very minor suggestion. Table 1 has been reordered chronologically (which I suggested and agree with), and the title was reworded to "Results of the literature review in order by appearance." The last part of that title was unclear to me until I read the authors' response to my review. Perhaps changing the wording to "ordered by publication date" or "ordered chronologically" might be better. Or just retaining the previous title, "Results of the literature review" would also be acceptable.

Author Response

Dear Reviewer,

Thank you very much for your positive feedback and for your thoughtful suggestion regarding the title of Table 1.

We have updated the title according to your recommendation, and it now reads: "Table 1. Results of the literature review ordered chronologically"

Kind regards,

M. Asuncion Vicente